# Liquid-Based Diagnostic Panels for Prostate Cancer: The Synergistic Role of Soluble PD-L1, PD-1, and mRNA Biomarkers

**DOI:** 10.3390/ijms26020704

**Published:** 2025-01-15

**Authors:** Margarita Žvirblė, Ieva Vaicekauskaitė, Žilvinas Survila, Paulius Bosas, Neringa Dobrovolskienė, Agata Mlynska, Rasa Sabaliauskaitė, Vita Pašukonienė

**Affiliations:** 1National Cancer Institute, P. Baublio Str. 3B, LT-08406 Vilnius, Lithuaniapaulius.bosas@nvi.lt (P.B.); neringa.dobrovolskiene@nvi.lt (N.D.); agata.mlynska@nvi.lt (A.M.); rasa.sabaliauskaite@nvi.lt (R.S.); vita.pasukoniene@nvi.lt (V.P.); 2Institute of Biosciences, Life Sciences Center Vilnius University, Saulėtekio av 7, LT-10257 Vilnius, Lithuania; zilvinas.survila@gmail.com; 3Vilnius Gediminas Technical University, Department of Chemistry and Bioengineering, Saulėtekio al 11, LT-10223 Vilnius, Lithuania

**Keywords:** prostate cancer, sPD-L1, sPD-1, mRNA transcripts, circulating molecules, liquid biopsy

## Abstract

This study aimed to evaluate the diagnostic potential of soluble Programmed Death Ligand 1 (sPD-L1) and Programmed Death 1 (sPD-1) molecules in plasma, along with urinary mRNA biomarkers—Prostate-Specific Membrane Antigen (*PSMA*), Prostate Cancer Antigen 3 (*PCA3*), and androgen receptor (*AR*) genes—for identifying clinically significant prostate cancer (PCa), defined as pathological stage 3. In a cohort of 68 PCa patients, sPD-L1 and sPD-1 levels were quantified using ELISA, while mRNA transcripts were measured by RT-qPCR. Results highlight the potential of integrating these liquid-based biomarkers. In particular, the combination of sPD-L1, sPD-1, and *AR* demonstrated the most significant improvement in diagnostic performance, increasing the area under the curve (AUC) from 0.65 to 0.81 and sensitivity from 60% to 88%, compared to *AR* alone. *PSMA* demonstrated an AUC of 0.82 and a specificity of 52.8%, which improved to an AUC of 0.85 and a specificity of 94.4% with the inclusion of sPD-L1 and sPD-1. Similarly, *PCA3* achieved an AUC of 0.75 and a specificity of 53.8%, increasing to an AUC of 0.78 and a specificity of 76.9% when combined with these biomarkers. Incorporating sPD-L1 into a three-gene panel further elevated the AUC from 0.74 to 0.94. These findings underscore the value of multimodal liquid-based diagnostic panels in improving the management of clinically significant PCa.

## 1. Introduction

Prostate cancer (PCa) remains the second most prevalent cancer in men globally [1,2], comprising roughly 15% of all cancer diagnoses worldwide. Forecasts indicate that the annual number of new prostate cancer cases is expected to increase from 1.4 million in 2020 to 2.9 million by 2040, based on analysis of global demographic shifts and the rising rates of life expectancy [3]. Considering that PCa is characterized as a heterogeneous disease [4], a variety of risk factors are involved in prostate cancer progression such as environmental, genetic and molecular factors [5]. Given this complexity, there is an urgent need for innovative, minimally invasive diagnostic strategies that prioritize precision oncology and personalized medicine. Liquid biopsy has recently gained significant attention as a promising tool in prostate cancer stratification, especially in genome sequencing profiling methods [6]. Meanwhile, liquid biopsy techniques that analyze soluble molecules by ELISA methods and urinary mRNA transcripts using RT-qPCR are well-suited for routine diagnostic applications. Compared to genome sequencing profiling, these methods provide enhanced scalability, cost-efficiency and economic feasibility. They also offer reliable predictive value and significantly shorter turnaround times, making them practical for widespread clinical implementation across diverse healthcare settings. Circulating plasma molecules such as soluble PD-L1 and PD-1 (sPD-L1 and sPD-1) have drawn notable focus in recent research due to their potential as prognostic and predictive markers in different cancer types [7,8,9] and demonstrated prognostic significance in our previous research on prostate cancer [10]. Urinary mRNAs, including those of *PSMA, PCA3*, and *AR*, play a significant role in prostate cancer development. These biomarkers provide valuable insights into the genetic landscape of tumors and are widely utilized in the diagnosis of prostate cancer [11,12,13,14,15,16,17,18]. The novel approach of combining multiple biomarkers has the potential to improve the accuracy of PCa detection and risk stratification. By integrating data from diverse biomarkers, a comprehensive immune and molecular profile of the patient’s disease can be constructed, providing valuable insights into the prediction of clinically significant prostate cancer. This approach underscores the expanded utility of liquid biopsies in future clinical applications.

## 2. Results

### 2.1. Biomarker Association with Prostate Cancer Clinical Features

Analysis of relative *AR*, *PCA3*, and *PSMA* mRNA expression in urine discovered a significant increase in *PSMA* (*p* ≤ 0.001) and *PCA3* (*p* ≤ 0.05) expression in clinically significant PCa when compared with clinically insignificant PCa cases, if classified as pathological stage pT3 (Figure 1 and Appendix A), as well as significant associations between *PSMA* expression and tumor grade (grade 1 vs. grade 3 *p* = 0.005, grade 1 vs. grade 2 *p* = 0.011) (Appendix A).

Soluble PD-1 and PD-L1 revealed sPD-L1 association with clinically significant PCa (sPDL1 *p* = 0.031) (Appendix A), and grade 3 PCa (grade 2 vs. grade 3 sPDL1 = 0.026) (Appendix A), while sPD-1 showed no differences in any of the clinical features examined.

No significant association between relative *AR*, *PCA3*, and *PSMA* mRNA expression and either the plasma biomarkers (sPD-L1 or sPD-1) or other clinical features (age, serum PSA concentration or immune cell count) was discovered.

### 2.2. Prediction of Clinically Significant PCa Using Liquid Biopsy Biomarkers

ROC analysis revealed *PSMA* to be the best single gene expression biomarker predictor of clinically significant PCa (AUC = 0.82) (Figure 2) with the highest sensitivity (1.00). On the other hand, sPD-1 showed the best single biomarker specificity (0.85), but lowest sensitivity (0.44).

Regarding the combination of urine and plasma biomarkers together, an increase in AUC values was noticed. While combining the three mRNAs’ expression did not increase the prediction of clinically significant PCa (AUC 0.74 vs. AUC 0.82 of *PSMA* and AUC 0.75 of *PCA3* biomarker), the panel comprising three mRNA transcripts along with sPD-L1 demonstrates significant enhancement in diagnostic properties compared to mRNA transcripts without sPD-L1, with an AUC of 0.94, accuracy of 0.96, sensitivity of 0.83 and specificity of 100%. The inclusion of sPD-1 did not enhance the diagnostic performance of the three-gene panel (Figure 3).

Of note, the addition of plasma sPD-L1 and sPD-1 to *PSMA* and *PCA3* gene expression biomarkers slightly increased AUC by (0.82 to 0.85) and (0.75 to 0.78), respectively, compared to the biomarkers alone. The combination of *AR* and two plasma biomarkers overall showed the most substantial improvement (AUC 0.65 to 0.81), compared to a single mRNA alone, in separation of clinically significant PCa out of all biomarkers examined.

While *PSMA* emerged as the most robust standalone biomarker (AUC 0.82), *AR* demonstrated modest performance (AUC 0.65). However, the combination of *AR* mRNA from urine with sPD-L1 and sPD-1 significantly improved diagnostic accuracy, increasing AUC from 0.65 to 0.81 and sensitivity from 60% to 88% (Figure 2 and Figure 4). This combination achieved comparable performance to the best *PSMA* + sPD-L1 + sPD-1 combination (AUC 0.81 vs. 0.85) and outperformed the combination with *PCA3* (AUC 0.78) (Figure 4).

## 3. Discussion

### 3.1. Significance of sPD-L1 and sPD-1 Along with mRNA of PSMA, PCA3 and AR Genes in PCa

In the context of intensive investigations on convenient biomarkers, a novel multifactorial approach that combines urine and blood biomarkers encompassing various aspects of the disease not only enhances detection but also offers a comprehensive assessment of prostate cancer. This approach highlights the potential of non-invasive liquid biopsies in improving the diagnosis and management of PCa. Building on our previous research, which identified plasma sPD-L1/sPD-1 as a potential biomarker of PCa [10], we investigated gene expression in the urine samples of the same patients. As shown in Figure 1 and Appendix A, sPD-L1 can differentiate between clinically significant and non-significant prostate cancer and is associated with higher tumor stages (*p* = 0.031) and ISUP grading (*p* = 0.026) in PCa. Similarly, elevated sPD-L1 levels are consistently linked to larger tumors, advanced stages, and metastasis across different cancers [19,20].

In our study, significant associations were identified between *PSMA* (*p* ≤ 0.001) and *PCA3* (*p* ≤ 0.05) expression and clinically significant prostate cancer (Figure 1 and Appendix A), and among the three genes examined, *PSMA* emerged as the most reliable single biomarker for predicting clinically significant PCa with an AUC of 0.82 (Figure 2). Similarly, Rigau reported that *PSMA* (AUC 0.74) outperformed *PSGR* (AUC 0.66) and *PCA3* (AUC 0.61) in predicting PCa within the PSA “gray zone” of 4–10 ng/mL [21]. Furthermore, *PSMA* was also linked to ISUP grading (Figure 1 and Appendix A), indicating its potential as a biomarker for disease severity and progression. Despite *AR* not demonstrating any association with cancer advancement, in single-biomarker assessment, it exhibited higher diagnostic accuracy (0.76 vs. 0.59) and specificity (0.81 vs. 0.48) than all three urine biomarkers combined (Figure 2 and Figure 3). Comparative analysis with other studies also suggests the involvement of *PSMA*, *PCA3*, and *AR* genes in prognosis and prediction of PCa. Blood PSMA-based biomarkers have been linked to malignancy risk [22] and predicted worse survival rates in metastatic PCa [23]. Higher PSMA expression correlated with advanced tumor stages and grades in biopsies and prostatectomy specimens [24]. Urine exosomal PSMA showed high diagnostic accuracy for significant PCa, correlating strongly with Gleason scores [25]. Similarly, *PCA3* scores have been associated with tumor aggressiveness [26], higher Gleason scores [15,27] and advanced clinical stages [27]. Moreover, various non-coding RNAs have been shown to influence prostate cancer progression by modulating *AR* signaling, highlighting their potential as biomarkers and therapeutic targets [28].

Although studies have demonstrated the utility of monitoring RNA transcripts from *PSMA*, *PCA3*, and *AR* genes for prostate cancer diagnosis, relying solely on these biomarkers may be limiting. These biomarkers, while associated with disease progression, may not fully capture the multifaceted nature of prostate cancer, including its clinical heterogeneity. As a result, important aspects of the disease, such as its diverse pathways, could remain undetected.

### 3.2. Combinations of Plasma sPD-L1/sPD-1 with mRNA of PSMA, PCA3 and AR Genes in PCa

The combination of several different biomarkers has been shown to be a promising approach to improve PCa diagnosis [29,30]. In our study, the combination of all three mRNA expressions did not improve the prediction of clinically significant prostate cancer (AUC 0.74) compared to using *PSMA* and *PCA3* alone (AUC 0.82 and 0.75, respectively) (Figure 2 and Figure 3). However, adding sPD-L1 to the triple gene expression panel significantly enhanced the model’s performance. This resulted in a diagnostic accuracy of 0.96, an AUC of 0.94, and an increase in specificity from 48% to 100%, as shown in Figure 3. The composition of three genes along with sPD-1 did not improve the diagnostic efficacy (Figure 3). Such a multifaceted approach, combining mRNA expression analysis of *PCA3, PSMA*, and *AR* genes with sPD-L1 levels, provides a comprehensive understanding of prostate cancer’s characteristics. Together, *PSMA*, *PCA3*, and *AR*, which play pivotal roles in prostate cancer development, represent key aspects of tumor biology, such as cancer progression [12,14] and androgen-dependent growth [16,17]. When integrated with sPD-L1 levels, indicative of immune evasion [9], these biomarkers collectively may offer an in-depth perspective on tumor biology, immune response dynamics, and the heterogeneity of prostate cancer, supporting improved diagnostic and therapeutic strategies. sPD-L1 has emerged as a promising biomarker for various cancers, including gastric [31] and lung cancers [32]. These findings highlight sPD-L1’s broader applicability across cancers, making it valuable for diagnostics and treatment monitoring, due to its direct involvement in immune suppression [33], correlation with tumor burden, aggressiveness [34] and consistent association with clinical outcomes [8,35,36]. In contrast, sPD-1 primarily reflects immune activation; however, high pretherapeutic sPD-1 levels suggest worse prognosis [9,37]. Previous studies have described correlations rather than combinations involving sPD-L1 and sPD-1. sPD-L1 is linked to neutrophil-to-lymphocyte ratio in advanced cancers [36]. Higher levels are linked to low hemoglobin and albumin and elevated C-reactive protein in gastric cancer [38]. In pancreatic cancer, combining sPD-L1/PD-L2/B7-H5/CA19-9 improves diagnostic sensitivity, though sPD-1 did not add significance [39]. sPD-1 and sPD-L1 levels together indicated treatment outcomes in PD-1 blockade therapy as well [40,41].

The combination of three mRNA transcripts along with sPD-L1 demonstrated the strongest distinction of clinically significant PCa cases among all biomarker combinations tested (Figure 3). Furthermore, combination of urinary sediment mRNA and circulating molecules (*AR*, sPD-L1, sPD-1), showed the most significant improvement over other biomarkers, highlighting its potential; the mRNA of urinary *AR* provides insights into the androgen receptor pathway, which is implicated in PCa development and progression [42,43]. Meanwhile, plasma sPD-L1 and PD-1 levels potentially reflect the tumor’s immune microenvironment [9,37]. Androgen receptor signaling has been found to affect the expression of PD-L1 in prostate cancer, with AR activation linked to higher PD-L1 levels [44,45]. Additionally, scores for AR activity were significantly positively correlated with PD-1 methylation, resulting in an association with significantly reduced BCR (biochemical recurrence)-free survival after radical prostatectomy [46], suggesting an AR influence on the PD-L1/PD-1 axis. To analyze this promising combination in more detail, the influence of each soluble molecule was examined individually. Notably, while sPD-L1 and sPD-1 together, along with *PSMA* and *PCA3*, each showed a 3-unit increase in AUC (Figure 2 and Figure 4), even single sPD-1 combined with *AR* demonstrated the ability to increase AUC with a 5-unit improvement from 0.65 to 0.70 (Figure 4).

This study serves as an initial exploration of combining sPD-1 with a non-sPD-L1 biomarker across multiple cancers, demonstrating that incorporating sPD-1 has the potential to further enhance the diagnostic capabilities of the *AR* biomarker. A diagnostic panel combining *AR* with sPD-L1 and sPD-1 may offer potential economic benefits and practical applicability for implementation. These findings require further investigation in larger cohorts and hold the potential to advance understanding of the mechanistic interplay between *AR* and sPD-L1/sPD-1 signaling. Integrating blood and urine biomarkers together significantly improves PCa detection and is supported by commercially available tests. SelectMDx Urine Test, including *DLX1*, *HOXC6*, *KLK3(PSA)* and other parameters achieved an AUC of 0.85 with 93% sensitivity and 47% specificity [47], while the Michigan Prostate Score (MiPS), consisting of urine mRNA of *T2-ERG* and *PCA3* with serum PSA also outperformed the regular PSA test [48]. Additionally, scientific studies confirm the effectiveness of combining biomarkers obtained from different body fluids. Urinary exosomal *PCA3* and *PSMA* with serum PSA and PI-RADS achieved higher AUC than PSA alone [49], as well as urinary *PCA3* enhanced diagnostic performance of PSA in high-risk populations [50].

## 4. Materials and Methods

### 4.1. Characteristics of PCa Population

In a cohort of PCa patients evaluated for soluble PD-L1 and PD-1 levels in our previous research [10], gene expression was additionally examined in 72 cases to further assess their diagnostic value in this study. Four cases were removed due to outlier values in gene expression; thus, the study included 68 PCa patients. The PCa cases were divided into clinically significant and not clinically significant PCa groups where clinical significance was defined as cases with pathological stage pT3, following the high-risk prostate cancer criteria summarized in the review by Wilkins [51]. These patients were deemed to have an unfavorable PCa risk. The clinical characteristics of the PCa group are provided in Table 1.

The inclusion and exclusion criteria are well described in the paper by Bosas, clearly outlining the participant selection process [52].

### 4.2. Blood Sampling

The blood sampling of sPD-L1 and sPD-1 is thoroughly detailed in our previous paper [10].

### 4.3. Urine Sampling

The urine sampling is described in detail in previous research [53].

### 4.4. Analysis of Soluble PD-L1 and PD-1

Commercially available ELISA kits for PD-L1 and PD-1 were used to measure the soluble forms of both proteins in plasma, following the manufacturer’s instructions (Invitrogen, Thermo Fisher Scientific, Vienna, Austria). sPD-L1 and sPD-1 control samples were included in each kit at known concentrations. The optical density was measured using plate reader BioTek Elx800 TM (BIO-Tek Instruments, Inc., Winooski, VT, USA) at 450 nm. Two duplicates of each sample were measured. Blanks and standards were assayed as directed by manufacturers.

### 4.5. Analysis of mRNA Expression of PCA3, PSMA and AR Genes

Total RNA from washed urine sediment samples was extracted using the TRIzol Reagent (Invitrogen, Thermo Fisher Scientific (TFS), Carsbad, CA, USA) following the manufacturer’s protocol. The RNA concentration and purity were assessed using a Nanodrop 2000 spectrophotometer (Thermo Scientific, Wilmington, DE, USA). The RNA samples were stored at −80 °C until the copy DNA (cDNA) synthesis step. Two-Step RT-qPCR was used to assay AR, PSMA, and PCA3 mRNA relative quantities in the urine sediment samples. The Maxima First Stand cDNA Synthesis Kit for RT-qPCR with dsDNase (TFS, Vilnius, Lithuania) and the Maxima SYBR Green qPCR Master Mix (2X), with separate ROX vial (TFS, Vilnius, Lithuania), were used for two-step RT-qPCR following the manufacturer’s protocols. The qPCR reactions were performed on QuantStudio 5 Real-Time PCR System (Applied Biosystems, TFS, Singapore). RT-qPCR data pre-processing was performed on QuantStudio Design & Analysis software v1.4.3 (Applied biosystems, TFS, Singapore). The quantification cycle (Ct) values were reported using the automatic threshold baseline. Ct values <35 cycles were removed from subsequent analysis. For each sample, melt–curve analysis was performed to evaluate the amplicon size. The initial Ct values normalized to the *HPRT1* gene expression using log2_2_^−ΔCt^ and then divided by the *PSA* gene expression; these normalized relative expression values were used in further statistical data analysis.

### 4.6. Statistical Analysis

Statistical analysis and data visualization was performed on Python version 3.11.5 (Python Software Foundation) and R version 4.3.1 [54,55] software. Data normalcy was determined using Shapiro–Wilk W tests. Cases exceeding three interquartile ranges were deemed outliers and removed from all statistical analysis. Associations between two independent samples were tested using Welch’s *t* test or Mann–Whitney U test as appropriate. Receiver operating characteristic curve (ROC) analysis [56] together with logistic regression was utilized to measure biomarker and feature combination accuracy to predict clinically significant PCa. In each ROC curve, continuous biomarker values were used in the analysis. For each ROC curve, the best threshold value was determined using the Youden method. Results were considered significant when *p* ≤ 0.05.

## 5. Conclusions

Our results demonstrate that the combination of multiple biomarkers presents new opportunities for liquid biopsies to identify effective combinations that reflect the multifactorial nature of the disease. The inclusion of plasma sPD-L1 and sPD-1 in a diagnostic panel, together with urinary *PSMA*, *PCA3*, and *AR* mRNA transcripts, has the potential to significantly improve PCa diagnostics. Urine and plasma are easily accessible biofluids, allowing for less invasive and repeatable sampling and longitudinal monitoring and potentially reducing unnecessary biopsies [57]. Future efforts should focus on refining multimodal liquid-based data panels for more precise cancer management, while also emphasizing the adoption of commonly used, cost-effective laboratory methods to improve accessibility.

## Figures and Tables

**Figure 1 ijms-26-00704-f001:**
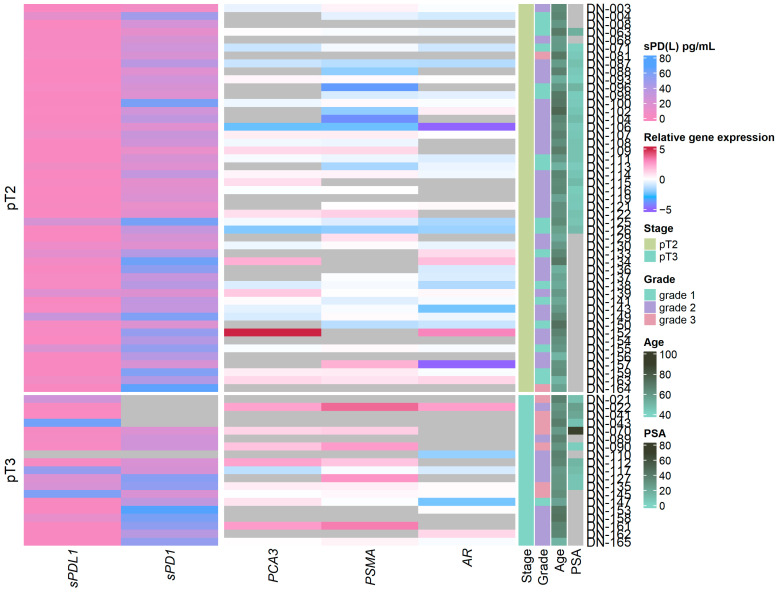
Heatmap depicting sPD-L1/sPD-1 biomarker concentrations in plasma and gene expression in urine sediment samples from prostate cancer patients together with clinical features. Clinically significant cases: pathological stage pT3. ISUP grade 1, 2 or 3. Grey color depicts no available data.

**Figure 2 ijms-26-00704-f002:**
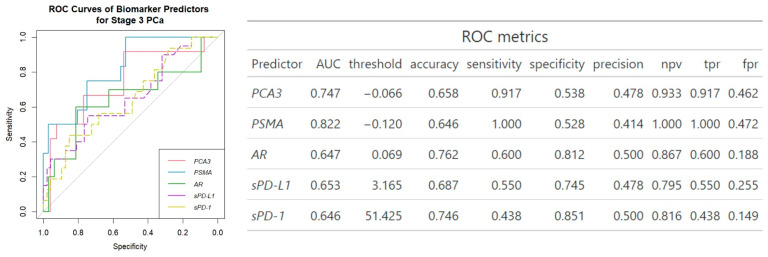
ROC analysis of biomarker prediction of clinically significant PCa. npv—negative predictive value, tpr—true positive rate, fpr—false positive rate.

**Figure 3 ijms-26-00704-f003:**
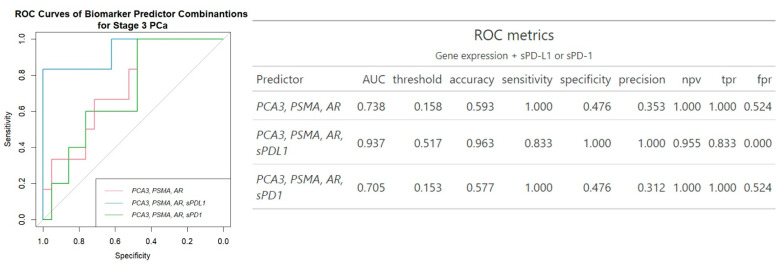
ROC analysis of biomarker combinations for prediction of clinically significant PCa. Npv—negative predictive value, tpr—true positive rate, fpr—false positive rate.

**Figure 4 ijms-26-00704-f004:**
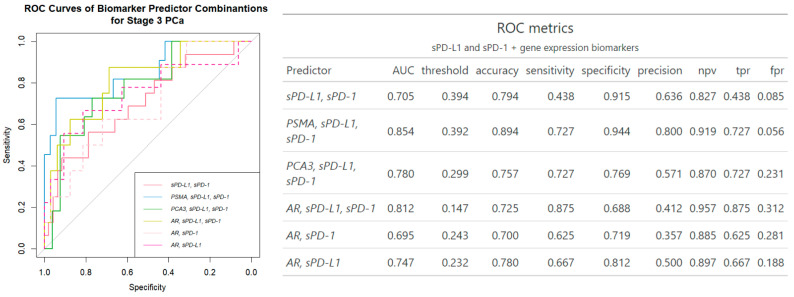
ROC analysis of single gene expression and serum biomarker combinations for prediction of clinically significant PCa. Npv—negative predictive value, tpr—true positive rate, fpr—false positive rate.

**Table 1 ijms-26-00704-t001:** Clinical characteristics of the PCa group.

Clinical Characteristic	Clinically not Significant PCa	Clinically Significant PCa	All Cases	*p*
n =	47	21	68	-
Mean Age (min–max)	67.7 (56–82)	69.2 (56–78)	68.2 (56–82)	0.33
Median PSA (pre-op) (IQR)	6.00 (3.52)	7.78 (8.80)	6.23 (4.10)	0.11
ISUP grade:				
ISUP 1	17	1	18	<0.001
ISUP 2	28	13	41	
ISUP 3	2	7	9	
Stage:				
pT2	47	-	47	0.001
pT3	-	21	21	

## Data Availability

The data presented in this study are available on request from the corresponding author due to protection of participant’s privacy.

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
