# Peer review of "Liquid-Based Diagnostic Panels for Prostate Cancer: The Synergistic Role of Soluble PD-L1, PD-1, and mRNA Biomarkers"

_ijms, 2025, doi:10.3390/ijms26020704_

Round 1

Reviewer 1 Report

Comments and Suggestions for Authors

The authors have conducted a study to evaluate the discriminatory ability of a biomarker combination in distinguishing clinically significant from clinically insignificant prostate cancer. The analysis could be interesting, but I have multiple concerns about the methodology and conclusions. Details below by row number:

12 – ‘circulating’ is unclear, would replace with ‘plasma’ ; sPD-L1, sPD-1 are abbreviations for ‘soluble PD-L1’ etc and this should be specified.

13 – all abbreviations should be explicated ‘PD-L1’ = programmed death ligand 1, AR = androgen receptor etc

15 – ‘Our findings…’ is a conclusive statement which comes before any results have been presented; this sentence is also nearly 6 lines. Suggest rephrasing for clarity and breaking up into shorter sentences.

20 – Same issue with length and complexity for the sentences starting on row 20. Would probably make more sense to present results with single biomarkers and then present how the test parameters are improved by consideration of multiparametric testing.

24 – the conclusion is misleading as no data has been presented on the actual management of prostate cancer; nor has the definition of clinically relevant prostate cancer been clarified at this stage; finally, in what way are these methods ‘easily applicable’? Are they CLIA standardized? Are they high throughput? This does not appear to be the case.

Overall, the abstract could be rewritten and improved for clarity andtr to only include claims supported by the analysis.

37 – Although the sentence is factually true, reference [5] does not support it.

37-39 – Language could be improved for clarity. I suggest a native English speaker/writer to re-edit the manuscript.

41 -  The use of CTCs has fallen out of favor for prostate cancer due to the limited clinical guidance the test provides. In any case, the current study does not use CTCs, so the reference here does not appear to be very relevant.

47-49 – Sentence is unclear, needs restructuring.

52-56 – “Characterization” is a vague word. I suggest “risk stratification” as a more specific alternative. The final sentences of the introduction make claims about the uses of the combined biomarker liquid biopsy assay as for its use in diagnosis and treatment selection etc, whereas the only true conclusion supported by the analysis is that of these biomarker combinations potentially predicting clinically significant prostate cancer. This requires revision to suit the actual aims of the study.

Table 1.

It is notable that none of the patients had ISUP grades > 3. Thus any findings would not be generalizable to patients with higher grade tumors.

Table 1 notes median PSA 6.23 with IQR of 4.10 for the overall population. The heatmap, however, has a PSA scale ranging from 0 to 80 with most subjects suggested to have had PSA in the 20-40s based on the shade of grey. Why this discrepancy?

! It would be important to show data on how the biomarkers correlate to PSA concentration, since the purpose of this biomarker assay would presumably be to improve on PSA. Just by glancing at the heatmap, there is a suggestion that lower sPD-1 expression correlates with PSA.

The question also becomes how were these cases detected? PSA-screening? And what was their pre-op stage, ie is the biomarker assay able to accurately upstage/downstage a case?

121-132 This description of other ‘losing’ biomarker combinations is confusing and of unclear value. Would consider shortening or removing it.

168-171 Phrase is confusing and meaning is unclear. Suggest rewording.

173-174 The statement has no reference. No such combination is standard at this time.

174-176 The sentence is in a “While.., then…” format, but lacks a “then…”

180-183 This is speculative, but could be interesting if better defined and supported with citations

212-218 The language here needs to be softened. The sample of patients is small and does not include high Gleason grades. There is also no validation cohort and overfitting is a big risk here.

Methods. The methods section should be more detailed with regards to how the biomarker analyte values were used for generation of the ROC curves. Were there continuous variables used or cut-offs? And, if the latter, how were these chosen? Is there a PSMA value above which NPV and TPR are 100%? This would be clinically important. Also, none of the sentences have verbs which should be amended for easier reading.

There is no limitations section, which would be highly encouraged here.

Comments on the Quality of English Language

The manuscript will benefit tremendously from proofing by an English language editor.

Author Response

Dear Reviewer,

Thank you for your thorough review of this manuscript and for providing valuable insights to enhance its professionalism.

Please find our responses to your comments below. All corrections made in the manuscript based on your suggestions are highlighted in green.

Comments 1: 12 – ‘circulating’ is unclear, would replace with ‘plasma’ ; sPD-L1, sPD-1 are abbreviations for ‘soluble PD-L1’ etc and this should be specified.

Response 1: Thank you for pointing this out. We agree with this comment. “Circulating” is replaced with “molecules in plasma”. Line 13. All abbreviations in the abstract have been clarified. Lines 12-15; 20.

Comments 2: 13 – all abbreviations should be explicated ‘PD-L1’ = programmed death ligand 1, AR = androgen receptor etc

Response 2. Thank you for pointing this out. We agree with this comment. All abbreviations in the abstract have been clarified. Lines 12-15; 20.

Comments 3. 15 – ‘Our findings…’ is a conclusive statement which comes before any results have been presented; this sentence is also nearly 6 lines. Suggest rephrasing for clarity and breaking up into shorter sentences.

Response 3. Thank you for highlighting this, we agree with this comment. The statement beginning with "Our findings..." was replaced with more precise terminology and restructured into shorter, clearer sentences for improved readability and logical flow. Lines 17-24.

Comments 4. 20 – Same issue with length and complexity for the sentences starting on row 20. Would probably make more sense to present results with single biomarkers and then present how the test parameters are improved by consideration of multiparametric testing.

Response 4. The sentences were restructured into shorter, clearer statements. Standalone biomarker results are presented first, emphasizing their baseline performance, followed by results with the inclusion of sPD-L1, to demonstrate the improvements achieved through multiparametric testing. Lines 21-24.

Comments 5. 24 – the conclusion is misleading as no data has been presented on the actual management of prostate cancer; nor has the definition of clinically relevant prostate cancer been clarified at this stage; finally, in what way are these methods ‘easily applicable’? Are they CLIA standardized? Are they high throughput? This does not appear to be the case.

Response 5.Thank you for highlighting this, we agree with this comment. The final conclusion has been revised to align with our findings, including only claims supported by the analysis. Row 25-26. Additionally, the definition of clinically significant prostate cancer (PCa) has been explicitly included to ensure clarity and contextual relevance. Lines 15-16.

Comments 6. Overall, the abstract could be rewritten and improved for clarity andtr to only include claims supported by the analysis.

Response 6. We sincerely appreciate your insightful comments on the abstract. The abstract has been revised comprehensively to address all comments, ensuring clarity, coherence, and alignment with the feedback provided. Lines 12-26.

Comments 7. 37 – Although the sentence is factually true, reference [5] does not support it.

Response 7.The claims in this sentence were originally derived from Table 4 of the previously cited article (PMID: 34017579); however, the reference has been updated to a more appropriate source for accuracy and relevance. Row 355 (in the list of References)

Comments 8. 37-39 – Language could be improved for clarity. I suggest a native English speaker/writer to re-edit the manuscript.

Response 8. Thank you for pointing this out. We sincerely appreciate your valuable feedback in enhancing the language of this manuscript. The sentence has been refined to enhance clarity. We have critically reviewed our manuscript for English language accuracy and made the necessary corrections. Additionally, our paper was reviewed by a colleague proficient in English to enhance linguistic precision and professionalism. Lines 37-39.

Comments 9. 41 -  The use of CTCs has fallen out of favor for prostate cancer due to the limited clinical guidance the test provides. In any case, the current study does not use CTCs, so the reference here does not appear to be very relevant.

Response 9. We replaced the reference with a more appropriate one based on genome sequencing profiling methods. Row 41 in the main text and row 357—359 in the list of References. Additionally, all advantages of techniques measuring sPD-L1/sPD-1 through ELISA and urinary mRNA transcripts via RT-qPCR were highlighted in comparison to genome sequencing profiling methods, which are currently regarded as leading approaches in the field of liquid biopsies. Lines 41-46.

Comments 10. 47-49 – Sentence is unclear, needs restructuring.

Response 10.  We agree with this suggestion. The sentence has been restructured to enhance clarity and readability. Lines 50-53.

Comments 11. 52-56 – “Characterization” is a vague word. I suggest “risk stratification” as a more specific alternative. The final sentences of the introduction make claims about the uses of the combined biomarker liquid biopsy assay as for its use in diagnosis and treatment selection etc, whereas the only true conclusion supported by the analysis is that of these biomarker combinations potentially predicting clinically significant prostate cancer. This requires revision to suit the actual aims of the study.

Response 11. We agree with this suggestion. The term "characterization" was replaced with the suggested term "risk stratification." Row 54. The statement was refined to reflect only the study's actual findings. Lines 55-57.

Table 1.

Comments 12. It is notable that none of the patients had ISUP grades > 3. Thus any findings would not be generalizable to patients with higher grade tumors.

Response 12. Thank you for highlighting this.We agree with this suggestion. This fact has been included as part of the identified limitations. Lines 307-310.

Comments 13. Table 1 notes median PSA 6.23 with IQR of 4.10 for the overall population. The heatmap, however, has a PSA scale ranging from 0 to 80 with most subjects suggested to have had PSA in the 20-40s based on the shade of grey. Why this discrepancy?

Response 13. Thank you for highlighting this. We sincerely apologize for this confusion. We have corrected the heatmap to include a wider range of colors to more accurately represent the PSA and age range as the grey color indicates missing values rather than mid-scale values as stated in the figure caption. The heatmap is displayed in Figure 1.

Comments 14! It would be important to show data on how the biomarkers correlate to PSA concentration, since the purpose of this biomarker assay would presumably be to improve on PSA. Just by glancing at the heatmap, there is a suggestion that lower sPD-1 expression correlates with PSA.

Response 14. As stated in lines 79-81 of the manuscript, there was no significant correlation between biomarker values and PSA concentrations. Please kindly find the attached data below.

Comments 15. The question also becomes how were these cases detected? PSA-screening? And what was their pre-op stage, ie is the biomarker assay able to accurately upstage/downstage a case?

Response 15. The cases in our cohort were detected via the national PSA screening program, followed by DRE and further testing with imaging. All clinical features included in our cohort were only taken at the date of diagnosis including PSA, grade and stage, thus we have ended up with some PSA values missing. Due to missing PSA values, we did not feel that comparison of our biomarkers with the PSA would be correct. However, we could make the comparison here. Taking PSA 10 ng/mL as a cutoff, same ROC analysis shows that the AUC for PSA in our cohort (25 cases with PSA <10 and 10 cases with PSA >10) did separate the pT3 cases with AUC 0.62, sensitivity 0.45, specificity 0.79. Thus, we could conclude that all biomarkers and biomarker combinations did accurately upstage and outperform the serum PSA as a biomarker, at least in AUC. We could not make the same comparison with the PSA cutoff of 4 ng/mL as only two of the cases in our cohort had a value of <4ng/mL Ultimately, according to [https://uroweb.org/guidelines/prostate-cancer] guidelines, serum PSA test is likely to detect <pT3 stage PCa cases,for which it is not recommended as a sole screening tool, without DRE or other tests.

Comments 16. 121-132 This descrption of other ‘losing’ biomarker combinations I  s confusing and of unclear value. Would consider shortening or removing it.

Response 16. Thank you for pointing this out. The paragraph was shortened to include only the most relevant results, aiming to avoid any potential confusion. Line 121-126.

Comments 17. 168-171 Phrase is confusing and meaning is unclear. Suggest rewording.

Response 17. We agree with this suggestion. The statement has been restructured to enhance clarity and readability. Lines 162-167.

Comments 18. 173-174 The statement has no reference. No such combination is standard at this time.

Response 18. We agree with this suggestion. The references supporting these statements have been added. Lines 413-417 in the list of References.

Comments 19. 174-176 The sentence is in a “While.., then…” format, but lacks a “then…”

Response 19. Thank you for pointing this out. The sentences have been restructured to improve clarity and readability. Lines 170-175.

Comments 20. 180-183 This is speculative, but could be interesting if better defined and supported with citations.

Response 20. We agres with this suggestion. The statement has been expanded for enhanced clarity and supplemented with appropriate citations. Lines 177-184.

Comments 21. 212-218 The language here needs to be softened. The sample of patients is small and does not include high Gleason grades. There is also no validation cohort and overfitting is a big risk here.

Response 21. Thank you for highlighting this. We agree with this suggestion. The language has been refined to adopt a less categorical tone. Lines 214-220. These facts have been included as part of the identified limitations. Row 307-310.

Comments 22A) Methods. The methods section should be more detailed with regards to how the biomarker analyte values were used for generation of the ROC curves. Were there continuous variables used or cut-offs? And, if the latter, how were these chosen?

Response 22A) Added in lines 285-286: In each ROC curve continuous biomarker values were used in the analyses. For each ROC the best threshold value was determined using the Youden method.

Commments 22B)  Is there a PSMA value above which NPV and TPR are 100%? This would be clinically important.

Response 22B). In case of PSMA gene expression and combinations of all three gene expression biomarker combinations and a combination of all three gene expression + sPDL1 serum concentration perfect values of negative predictive value (NPV) and true positive rate (TPR) were reached, showing the ability of these tests to correctly identify all pT3 cases and all pT2 cases. This provides a possible screening utility to such biomarker test as patients with biomarker values below the threshold would not need further testing for clinically advanced prostate cancer, while simultaneously detecting all pT3 prostate cancer cases. However, this result should be validated in a larger cohort as NPV might increase with low positive case count such as in our case. This fact has been included as part of the identified limitations. Lines 307-310.

Comments 22C). Also, none of the sentences have verbs which should be amended for easier reading.

Response 22C). Thank you for pointing this out. Added verbs to each sentence without it. Highlighted in green, lines 278, 279, 281, 283.

Comments 23. There is no limitations section, which would be highly encouraged here.

Response 23.  Thank you for highlighting this. We agree with this suggestion. The limitations section has now been incorporated. Lines 307-310.

Comments 24. Comments on the Quality of English Language. The manuscript will benefit tremendously from proofing by an English language editor.

Response 24. We thank you for your valuable comments in improving the language of this manuscript. All comments regarding sentence structure have been resolved, the entire manuscript has been critically reviewed for English language accuracy. Additionally, our paper was reviewed by a colleague proficient in English to enhance linguistic precision and clarity. We sincerely apologize for any prior inaccuracies.

Reviewer 2 Report

Comments and Suggestions for Authors

This manuscript reported the efficacy of diagnosing prostatic cancer using circulating molecules sPD-L1 and sPD-1 in combination with urinary mRNA expression of PCA3, PSMA and AR genes in diagnosing clinically significant prostate cancer (PCa).  Their results highlight the potential of integrating these liquid-based biomarkers together, particularly the promising combination of soluble PD-L1, PD-1 and AR, which demonstrates the best enhancement in diagnostic performance, increasing the area under the curve (AUC) and improving sensitivity in distinguishing clinically significant PCa. There are minor questions for authors to comment:

1) The authors did not measure these biomarkers in the controls or patients with suspicious chronic prostatitis, which might have effects on the accuracy of diagnostic rate of PCa.

2) The authors should estimate the cost effectiveness by adding these serum test for increases of diagnostic accuracy of PCa.

Author Response

Dear Reviewer,
Thank you for the time you spent on reviewing this manuscript and for your valuable insights.

Please find our responses based on your comments below. Corrections in the manuscript based on your comments are highlighted in purple.

Comments 1. The authors did not measure these biomarkers in the controls or patients with suspicious chronic prostatitis, which might have effects on the accuracy of diagnostic rate of PCa.

Response 1. We fully agree that comparative measurements in controls or patients with suspected chronic prostatitis would enhance the diagnostic accuracy of PCa. Unfortunately, we do not have comparative measurements of urinary mRNA transcripts in the control group. However, in our previous research (Zvirble et al., 2024; PMID: 39055716 ), we analyzed the levels of soluble PD-L1 and PD-1 in the same cohort of prostate cancer patients and healthy males.

The control group was carefully selected based on medical history and drug usage and was designed to closely match the study population’s age profile. This approach aligns with findings on the peak concentration of sPD-L1 in healthy individuals: sPD-L1 is present in healthy humans and significantly increases with age. Studies have shown that individuals aged 51–70 years exhibit the highest levels of sPD-L1 (Chen et al., 2011; Wu et al., 2019). These considerations made our control group an ideal reference for comparing sPD-L1 levels between PCa patients and healthy men.

Our investigation demonstrated elevated levels of soluble PD-L1 and PD-1 in the plasma of PCa patients, both before and after surgical treatment, compared to healthy controls.

In this article this fact has been included as part of the identified limitations. Lines 307-310.

Comments 2 The authors should estimate the cost effectiveness by adding these serum test for increases of diagnostic accuracy of PCa.

Response 2. Thank you for pointing this out. We agree with this suggestion. The cost-effectiveness and other advantages of techniques measuring sPD-L1/sPD-1 through ELISA and urinary mRNA transcripts via RT-qPCR were highlighted in comparison to newly added genome sequencing profiling methods, which are currently considered leading approaches in the field of liquid biopsies. Lines 41-46.

Round 2

Reviewer 1 Report

Comments and Suggestions for Authors

The authors have addressed all comments and the manuscript has seen dramatic improvement.